# Tractable Learning for Complex Probability Queries

**Jessa Bekker, Jesse Davis**
KU Leuven, Belgium
`{jessa.bekker,jesse.davis}@cs.kuleuven.be`

**Arthur Choi, Adnan Darwiche, Guy Van den Broeck**
University of California, Los Angeles
`{aychoi,darwiche,guyvdb}@cs.ucla.edu`

## Abstract

Tractable learning aims to learn probabilistic models where inference is guaranteed to be efficient. However, the particular class of queries that is tractable depends on the model and underlying representation. Usually this class is MPE or conditional probabilities $\Pr(\mathbf{x}|\mathbf{y})$ for joint assignments $\mathbf{x}, \mathbf{y}$. We propose a tractable learner that guarantees efficient inference for a broader class of queries. It simultaneously learns a Markov network and its tractable circuit representation, in order to guarantee and measure tractability. Our approach differs from earlier work by using Sentential Decision Diagrams (SDD) as the tractable language instead of Arithmetic Circuits (AC). SDDs have desirable properties, which more general representations such as ACs lack, that enable basic primitives for Boolean circuit compilation. This allows us to support a broader class of complex probability queries, including counting, threshold, and parity, in polytime.

## 1 Introduction

Tractable learning [1] is a promising new machine learning paradigm that focuses on learning probability distributions that support efficient querying. It is motivated by the observation that while classical algorithms for learning Bayesian and Markov networks excel at fitting data, they ignore the cost of reasoning with the learned model. However, many applications, such as health-monitoring systems, require efficient and (guaranteed) accurate reasoning capabilities. Hence, new learning techniques are needed to support applications with these requirements.

Initially, tractable learning focused on the first model class recognized to be tractable: low-treewidth graphical models [2–5]. Recent advances in probabilistic inference exploit other properties of a model, including local structure [6] and exchangeability [7], which even scale to models that have high treewidth. In particular, the discovery of local structure led to *arithmetic circuits* (ACs) [8], which are a much more powerful representation of tractable probability distributions. In turn, this led to new tractable learners that targeted ACs to guarantee efficient inference [9, 10]. In this context, ACs with latent variables are sometimes called sum-product networks (SPNs) [11, 12]. Other tractable learners target exchangeable models [13, 14] or determinantal point processes [15].

There is a trade-off in tractable learning that is poorly understood and often ignored: tractability is not absolute, and always relative to a *class of queries* that the user is interested in. Existing approaches define tractability as the ability to efficiently compute most-probable explanations (MPE) or conditional probabilities $\Pr(\mathbf{x}|\mathbf{y})$ where $\mathbf{x}, \mathbf{y}$ are joint assignments to subsets of random variables. While these queries are indeed efficient on ACs, many other queries of interest are not. For example, computing partial MAP remains NP-hard on low-treewidth and AC models [16]. Similarly, various

decision [17, 18], monotonicity [19], and utility [20] queries remain (co-)NP-hard.[1] Perhaps the simplest query beyond the reach of tractable AC learners is for probabilities $\Pr(\phi|\psi)$, where $\phi, \psi$ are complex properties, such as counts, thresholds, comparison, and parity of sets of random variables. These properties naturally appear throughout the machine learning literature, for example, in neural nets [21], and in exchangeable [13] and statistical relational models [22]. We believe they have not been used to their full potential in the graphical models' world due to their intractability. We call these types of queries *complex probability queries*.

This paper pushes the boundaries of tractable learning by supporting more queries efficiently. While we currently lack any representation tractable for partial MAP, we do have all the machinery available to learn tractable models for complex probability queries. Their tractability is enabled by the weighted model counting (WMC) [6] encoding of graphical models and recent advances in compilation of Boolean functions into Sentential Decision Diagrams (SDDs) [23]. SDDs can be seen as a syntactic subset of ACs with more desirable properties, including the ability to (1) incrementally compile a Markov network, via a conjoin operator, (2) dynamically minimize the size and complexity of the representation, and (3) efficiently perform complex probability queries.

Our first contribution is a tractable learning algorithm for Markov networks with compact SDDs, following the outer loop of the successful ACMN learner [9] for ACs, that uses SDD primitives to modify the circuit during the Markov network structure search. Support for the complex queries listed above also means that these properties can be employed as features in the learned network. Second, we prove that complex symmetric probability queries over $n$ variables, as well as their extensions, run in time polynomial in $n$ and linear in the size of the learned SDD. Tighter complexity bounds are obtained for specific classes of queries. Finally, we illustrate these tractability properties in an empirical evaluation on four real-world data sets and four types of complex queries.

## 2 Background

### 2.1 Markov Networks

A *Markov network* or *Markov random field* compactly represents the joint distribution over a set of variables $\mathbf{X} = (X_1, X_2, \ldots, X_n)$ [24]. Markov networks are often represented as *log-linear models*, that is, an exponentiated weighted sum of features of the state $\mathbf{x}$ of variables $\mathbf{X}$: $\Pr(\mathbf{X} = \mathbf{x}) = \frac{1}{Z} \exp \sum_j w_j f_j(\mathbf{x})$. The $f_j(\mathbf{X})$ are real-valued functions of the state, $w_j$ is the weight associated with $f_j$, and $Z$ is the partition function. For discrete models, features are often Boolean functions; typically a conjunction of tests of the form $(X_i = x_i) \wedge \cdots \wedge (X_j = x_j)$. One is interested in performing certain inference tasks, such as computing the posterior marginals or most-likely state (MPE) given observations. In general, such tasks are intractable (#P- and NP-hard).

Learning Markov networks from data require estimating the weights of the features (parameter learning), and the features themselves (structure learning). We can learn the parameters by optimizing some convex objective function, which is typically the log-likelihood. Evaluation of this function and its gradient is in general intractable (#P-complete). Therefore, it is common to optimize an approximate objective, such as the pseudo-log-likelihood. The classical structure learning approach [24] is a greedy, top-down search. It starts with features over individual variables, and greedily searches for new features to add to the model from a set of candidate features, found by conjoining pairs of existing features. Other approaches convert local models into a global one [25]. To prevent overfitting, one puts a penalty on the complexity of the model (e.g., number of features).

### 2.2 Tractable Circuit Representations and Tractable Learning

Tractable circuit representations overcome the intractability of inference in Markov networks. Although we are not always guaranteed to find a compact tractable representation for every Markov network, in this paper we will guarantee their existence for the learned models.

**AC** *Arithmetic Circuits* (ACs) [8] are directed acyclic graphs whose leafs are inputs representing either indicator variables (to assign values to random variables), parameters (weights $w_j$) or constants. Figure 1c shows an example. ACs encode the partition function computation of a Markov network.

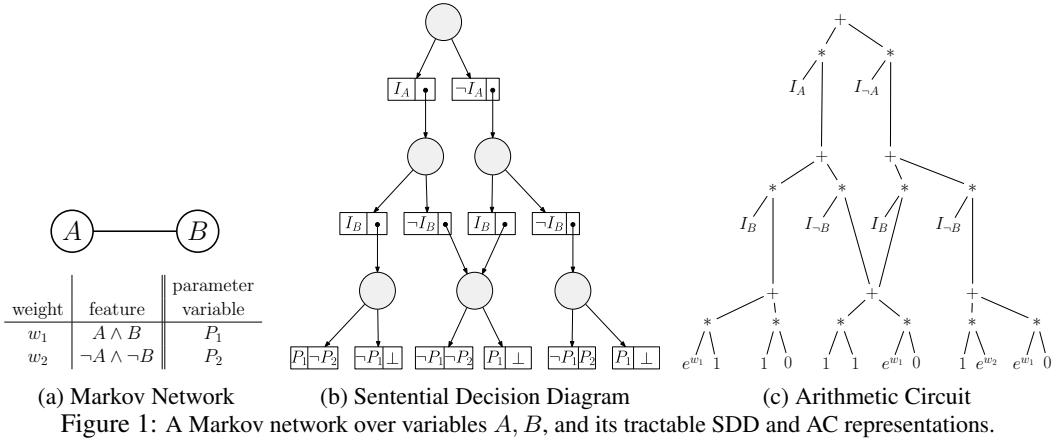

(a) Markov Network     (b) Sentential Decision Diagram     (c) Arithmetic Circuit

Figure 1: A Markov network over variables $A$, $B$, and its tractable SDD and AC representations.

By setting indicators to 1 and evaluating the AC bottom-up, the value of the partition function, $Z$, is obtained at the root. Other settings of the indicators encode arbitrary evidence. Moreover, a second, top-down pass yields all single-variable marginal probabilities; similar procedures exist for MPE. All these algorithms run in time linear in the size of the AC (number of edges). The *tractable learning* paradigm for Markov networks is best exemplified by ACMN [9], which concurrently learns a Markov network and its AC. It employs a complexity penalty based on the inference cost. Moreover, ACMN efficiently computes the exact log-likelihood (as opposed to pseudo-log-likelihood) and its gradient on the AC. ACMN uses the standard greedy top-down feature search outlined above.

**SDD** *Sentential Decision Diagrams* (SDDs) are a tractable representation of sentences in propositional logic [23]. The supplement[2] reviews SDDs in detail; a brief summary is next. SDDs are directed acyclic graphs, as depicted in Figure 1b. A circle node represents the disjunction of its children. A pair of boxes denotes the conjunction of the two boxes, and each box can be a (negated) Boolean variable or a reference to another SDD node. The detailed properties of SDDs yield two benefits. First, SDDs support an efficient conjoin operator that can incrementally construct new SDDs from smaller SDDs in linear time. Second, SDDs support dynamic minimization, which allows us to control the growth of an SDD during incremental construction.

There is a close connection between SDDs for logic and ACs for graphical models, through an intermediate *weighted model counting* formulation [6], which is reviewed in the supplement. Given a graphical model $M$, one can construct a logical sentence $\Delta$ whose satisfying assignments are in one-to-one correspondence with the possible worlds of $M$. Moreover, each satisfying assignment of $\Delta$ encodes the weights $w_j$ that apply to its possible world in $M$. For each feature $f_j$ of $M$, this $\Delta$ includes a constraint $f_j \Leftrightarrow P_j$, meaning that weight $w_j$ applies when "parameter" variable $P_j$ is true; see Figure 1a. A consequence of this correspondence is that, given an SDD for $\Delta$, we can efficiently construct an AC for the original Markov network $M$; see Figure 1. Hence, an SDD corresponding to $M$ is a tractable representation of $M$. Different from ACs, SDDs have the following properties: support for efficient (linear) conjunction allows us to add new features $f_j$ and incrementally learn a Markov network. Moreover, dynamic minimization lets us systematically search for more compact circuits for the same Markov network, mitigating the increasing complexity of inference as we learn more features. Such operations are not available for ACs in general.

## 3 Learning Algorithm

We propose LearnSDD, which employs a greedy, general-to-specific search that simultaneously learns a Markov network and its underlying SDD which is used for inference. The cost of inference in the learned model is dictated by the size of its SDD. Conceptually, our approach is similar to ACMN [9] with the key differences being our use of SDDs instead of ACs, which gives us more tractability and freedom in the types of features that are considered.

**Algorithm 1** LearnSDD($T, e, \alpha$)

---

   initialize model $M$ with variables as features
   $M_{best} \leftarrow M$
   **while** number of edges $|SDD_M| < e$ **and not** timeout
      best_score = $-\infty$
      $F \leftarrow$ generateFeatures($M, T$)
      **for** each feature $f$ in $F$ **do**
         $M' \leftarrow M$.add($f$)
         **if** score($M', T, \alpha$) $>$ best_score
            best_score = score($M', T, \alpha$)
            $M_{best} \leftarrow M'$
      $M \leftarrow M_{best}$

---

LearnSDD, outlined in Algorithm 1, receives as input a training set $T$, a maximum number of edges $e$, and a parameter $\alpha$ to control the relative importance of fitting the data vs. the cost of inference. As is typical with top-down approaches to structure learning [24], the initial model has one feature $X_i =$ true for each variable, which corresponds to a fully-factorized Markov network. Next, LearnSDD iteratively constructs a set of candidate features, where each feature is a logical formula. It scores each feature by compiling it into an SDD, conjoining the feature to the current model temporarily, and then evaluating the score of the model that includes the feature. The supplement shows how a features is added to an SDD. In each iteration, the highest scoring feature is selected and added to the model permanently. The process terminates when the maximum number of edges is reached or when it runs out of time. Inference time is dictated by the size of the learned SDD. To control this cost, we invoke dynamic SDD minimization each time a feature is evaluated, and when we permanently add a feature to the model.

Performing structure learning with SDDs offers advantages over ACs. First, SDDs support a practical conjoin operation, which greatly simplifies the design of a top-down structure learning algorithm (ACMN instead relies on a complex special-purpose AC modification algorithm). Second, SDDs support dynamic minimization, allowing us to search for smaller SDDs, as needed. The following two sections discuss the score function and feature generation in greater detail.

### 3.1 Score Function and Weight Learning

Score functions capture a trade-off between fitting the training data and the preference for simpler models, captured by a regularization term. In tractable learning, the regularization term reflects the cost of inference in the model. Therefore, we use the following score function:

$$\text{score}(M', T) = [\log \text{Pr}(T|M') - \log \text{Pr}(T|M)] - \alpha \left[ |SDD_{M'}| - |SDD_M| \right] / |SDD_M| \quad (1)$$

where $T$ is the training data, $M'$ is the model extended with feature $f$, $M$ is the old model, $|SDD_.|$ returns the number of edges in the SDD representation, and $\alpha$ is a user-defined parameter. The first term is the improvement in the model's log-likelihood due to incorporating $f$. The second term measures the relative growth of the SDD representation after incorporating $f$. We use the relative growth because adding a feature to a larger model adds many more edges than adding a feature to a smaller model. Section 4 shows that any query's inference complexity depends on the SDD size. Finally, $\alpha$ lets us control the trade-off between fitting the data and the cost of inference.

Scoring a model requires learning the weights associated with each feature. Because we use SDDs, we can efficiently compute the exact log-likelihood and its gradient using only two passes over the SDD. Therefore, we learn maximum-likelihood estimates of the weights.

### 3.2 Generating Features

In each iteration, LearnSDD constructs a set of candidate features using two different feature generators: conjunctive and mutex. The **conjunctive generator** considers each pair of features $f_1, f_2$ in the model and proposes four new candidates per pair: $f_1 \wedge f_2, \neg f_1 \wedge f_2, f_1 \wedge \neg f_2$ and $\neg f_1 \wedge \neg f_2$.

The **mutex generator** automatically identifies mutually exclusive sets of variables in the data and proposes a feature to capture this relationship. Mutual exclusivity arises naturally in data. It occurs in tractable learning because existing approaches typically assume Boolean data. Hence,

any multi-valued attribute is converted into multiple binary variables. For all variable sets $\mathbf{X} = \{X_1, X_2, \cdots, X_n\}$ that have exactly one "true" value in each training example, the *exactly one* feature $\bigvee_{i=1}^{n}(X_i \wedge \bigwedge_{j \neq i} \neg X_j)$ is added to the candidate set. When at most one variable is "true", the *mutual exclusivity* feature $\bigvee_{i=1}^{n}(X_i \wedge \bigwedge_{j \neq i} \neg X_j) \vee \bigwedge_{j=1}^{n} \neg X_j$ is added to the candidate set.

# 4  Complex Queries

Tractable learning focuses on learning models that can efficiently compute the probability of a query given some evidence, where both the query and evidence are conjunctions of literals. However, many other important and interesting queries do not conform to this structure, including the following:

- Consider the task of predicting the probability that a legislative bill will pass given that some lawmakers have already announced how they will vote. Answering this query requires estimating the probability that a *count* exceeds a given threshold.

- Imagine only observing the first couple of sentences of a long review, and wanting to assess the probability that the entire document has more positive words than negative words in it, which could serve as proxy for how positive (negative) the review is. Answering this requires *comparing two groups*, in this case positive words and negative words.

Table 1 lists these and other examples of what we call complex queries, which are logical functions that cannot be written as a conjunction of literals. Unfortunately, tractable models based on ACs are, in general, unable to answer these types of queries efficiently. We show that using a model with an SDD as the target tractable representation can permit efficient exact inference for certain classes of complex queries: symmetric queries and their generalizations. No known algorithm exists for efficiently answering these types of queries in ACs. For other classes of complex queries, the complexity is never worse than for ACs, and in many cases SDDs will be more efficient. Note that SPNs have the same complexity for answering queries as ACs since they are interchangeable [12].

We first discuss how to answer complex queries using ACs and SDDs. We then discuss some classes of complex queries and when we can guarantee tractable inference in SDDs.

## 4.1  Answering Complex Queries

Currently, it is only known how to solve conjunctive queries in ACs. Therefore, we will answer complex queries by asking multiple conjunctive queries. We convert the query into DNF format $\bigvee \mathbf{C}$ consisting of $n$ *mutually exclusive* clauses $\mathbf{C} = \{C_1, \ldots, C_n\}$. Now, the probability of the query is the sum of the probabilities of the clauses: $\Pr(\bigvee \mathbf{C}) = \sum_{i=1}^{n} \Pr(C_i)$. In the worst case, this construction requires $2^m$ clauses for queries over $m$ variables. The inference complexity for each clause on the AC is $O(|AC|)$. Hence, the total inference complexity is $O(2^m \cdot |AC|)$.

SDDs can answer complex queries without transforming them into mutually exclusive clauses. Instead, the query $Q$ can directly be conjoined with the weighted model counting formulation $\Delta$ of the Markov network $M$. Given an SDD $S_m$ for the Markov network and an SDD $S_q$ for $Q$, we can efficiently compile an SDD $S_a$ for $Q \wedge \Delta$. From $S_a$, we can compute the partition function of the Markov network after asserting $Q$, which gives us the probability of $Q$. This computation is performed efficiently on the AC that corresponds to $S_a$ (cf. Section 2.2). The supplement explains the protocol for answering a query. The size of the SDD $S_a$ is at most $|S_q| \cdot |S_m|$ [23], and inference is linear in the circuit size, therefore it is $O(|S_q| \cdot |S_m|)$. When converting an arbitrary query into SDD, the size may grow as large as $2^m$, with $m$ the number of variables in the query. But often it will be much smaller (see Section 4.2). Thus, the overall complexity is $O(2^m \cdot |S_m|)$, but often much better, depending on the query class.

## 4.2  Classes of Complex Queries

A first class of tractable queries are symmetric Boolean functions. These queries do not depend on the exact input values, but only on how many of them are true.

**Definition 1.** A Boolean function $f(X_1, \ldots, X_n) : \{0, 1\}^n \to \{0, 1\}$ is a *symmetric query* precisely when $f(X_1, \ldots, X_n) = f(X_{\pi(1)}, \ldots, X_{\pi(n)})$ for every permutation $\pi$ of the $n$ indexes.

Table 1: Examples of complex queries, with $m$ the SDD size and $n$ the number of query variables.

| Query class | Query Type | Inference Complexity | Example |
|---|---|---|---|
| Symmetric Query | Parity | $O(mn)$ | $\#(A, B, C)\%2 = 0$ |
| | $k$-Threshold | $O(mnk^2)$ | $\#(A, B, C) > 1$ |
| | Exactly-$k$ | $O(mnk^2)$ | $\#(A, B, C) = 2$ |
| | Modulo-$k$ | $O(mnk)$ | $\#(A, B, C)\%3 = 0$ |
| Asymmetric Tractable Query | Exactly-$k$ | $O(mnk^2)$ | $\#(A, B, \neg C) = 2$ |
| | Hamming distance $k$ | $O(mnk^2)$ | $\#(A, B, \neg C) \leq 2$ |
| | Group comparison | $O(mn^3)$ | $\#(A, B, \neg C) > \#(D, \neg E)$ |

Table 1 lists examples of functions that can always be answered in polytime because they have a compact SDD. Note that the features generated by the mutex generator are types of exactly-$k$ queries where $k = 1$, and therefore have a compact SDD. We have the following result.

**Theorem 1.** *Markov networks with compact SDDs support tractable querying of symmetric functions. More specifically, let $M$ be a Markov network with an SDD of size $m$, and let $Q$ be any symmetric function of $n$ variables. Then, $\mathrm{Pr}_M(Q)$ can be computed in $O(mn^3)$ time. Moreover, when $Q$ is a parity function, querying takes $O(mn)$ time, and when $Q$ is a $k$-threshold or exactly-$k$ function, querying takes $O(mnk^2)$ time.*

The proof shows that *any* SDD can be conjoined with these queries without increasing the SDD size by more than a factor polynomial in $n$. The proof of Theorem 1 is given in the supplement. This tractability result can be extended to certain non-symmetric functions. For example, negating the inputs to a symmetric functions still yields a tractable complex query. This allows queries for the probability that the state is within a given Hamming distance from a desired state. Moreover, Boolean combinations of a bounded number of tractable function also admit efficient querying. This allows queries that compare symmetric properties of different groups of variables.

We cannot guarantee tractability for other classes of complex queries, because some queries do not have a compact SDD representation. An example of such a query is the weighed $k-$threshold where each literal has a corresponding weight and the total weight of true literals must be bigger than some threshold. While the worst-case complexity of using SDDs and ACs to answer such queries is the same, we show in the supplement that SDDs can still be more efficient in practice.

## 5 Empirical Evaluation

The goal of this section is to evaluate the merits of using SDDs as a target representation in tractable learning for complex queries. Specifically, we want to address the following questions:

**Q1** Does capturing mutual exclusivity allow LearnSDD to learn more accurate models than ACMN?

**Q2** Do SDDs produced by LearnSDD answer complex queries faster than ACs learned by ACMN?

To resolve these questions, we run LearnSDD and ACMN on real-world data and compare their performance. Our LearnSDD implementation builds on the publicly available SDD package.[3]

### 5.1 Data

Table 2 describes the characteristics of each data set.

Table 2: Data Set Characteristics

| Data Set | Train Set Size | Tune Set Size | Test Set Size | Num. Vars. |
|---|---|---|---|---|
| Traffic | 3,311 | 441 | 662 | 128 |
| Temperature | 13,541 | 1,805 | 2,708 | 216 |
| Voting | 1,214 | 200 | 350 | 1,359 |
| Movies | 1,600 | 150 | 250 | 1000 |

**Mutex features** We used the Traffic and Temperature data sets [5] to evaluate the benefit of detecting mutual exclusivity. In the initial version of these data sets, each variable had four values, which were binarized using a 1-of-n encoding.

**Complex queries** To evaluate complex queries, we used voting data from `GovTrac.us` and Pang and Lee's Movie Review data set.[4] The voting data contains all 1764 votes in the House of Representatives from the 110th Congress. Each bill is an example and the variables are the votes of the 453 congressmen, which can be yes, no, or present. The movie review data contains 1000 positive and 1000 negative movie reviews. We first applied the Porter stemmer and then used the Scikit Learn CountVectorizer,[5] which counts all 1- and 2-grams, while omitting the standard Scikit Learn stop words. We selected the 1000 most frequent n-grams in the training data to serve as the features.

## 5.2 Methodology

For all data sets, we divided the data into a single train, tune, and test partition. All experiments were run on identically configured machines with 128GB RAM and twelve 2.4GHz cores.

**Mutex features** Using the training set, we learned models with both LearnSDD and ACMN. For LearnSDD, we tried setting $\alpha$ to 1.0, 0.1, 0.01 and 0.001. For ACMN, we did a grid search for the hyper-parameters (per-split penalty $ps$ and the L1 and L2-norm weights $l1$ and $l2$) with $ps \in \{2, 5, 10\}$, $l1 \in \{0.1, 1, 5\}$ and $l2 \in \{0.1, 0.5, 1\}$. For both methods, we stopped learning if the circuit exceeded two million edges or the algorithm ran for 72 hours. For each approach, we picked the best learned model according to the tuning set log-likelihood. We evaluated the quality of the selected model using the log-likelihood on the test set.

**Complex queries** In this experiment, the goal is to compare the time needed to answer a query in models learned by LearnSDD and ACMN. In both SDDs and ACs, inference time depends linearly on the number of edges in the circuit. Therefore, to ensure a fair comparison, the learned models should have approximately the same number of edges. Hence, we first learned an SDD and then used the number of edges in the learned SDD to limit the size of the model learned by ACMN.

In the voting data set, we evaluated the threshold query: what is the probability that at least 50% of the congressmen vote "yes" on a bill, given as evidence that some lawmakers have already announced their vote? We vary the percentage of unknown votes from 1 to 100% in intervals of 1% point. We evaluated several queries on the movie data set. The first two queries mimic an active sensing setting to predict features of the review without reading it entirely. The evidence for each query are the features that appear in the first 750 characters of the stemmed review. On average, the stemmed reviews have approximately 3,600 characters. The first query is $\Pr(\#(positive\ ngrams) > 5)$ and second is $\Pr(\#(positive\ ngrams) > \#(negative\ ngrams))$, which correspond to a threshold query and a group comparison query, respectively. For both queries, we varied the size of the positive and negative ngram sets from 5 to 100 ngrams with an increment size of 1. We randomly selected which ngrams are positive and negative as we are only interested in a query's evaluation time. The third query is the probability that a parity function over a set of features is even. We vary the number of variables considered by the parity function from 5 to 100. For each query, we report the average per example inference time for each learned model on the test set. We impose a 10 minute average time limit and 100 minutes individual time limit for each query. For completeness, the supplement reports run times for queries that are guaranteed to (not) be tractable for both ACs and SDDs as well as the conditional log-likelihoods of all queries.

## 5.3 Results and Discussion

**Mutex features** Figure 2 shows the test set log-likelihoods as a function of the size of the learned model. In both data sets, LearnSDD produces smaller models that have the same accuracy as AC. This is because it can add mutex features without the need to add other features that are needed as building blocks but are redundant afterwards. These results allow us to affirmatively answer **(Q1)**.
**Complex queries** Figure 3 shows the inference times for complex queries that are extensions of symmetric queries. For all queries, we see that LearnSDD's model results in significantly faster inference times than ACMN's model. In fact, ACMN's model exceeds the ten minute time limit on

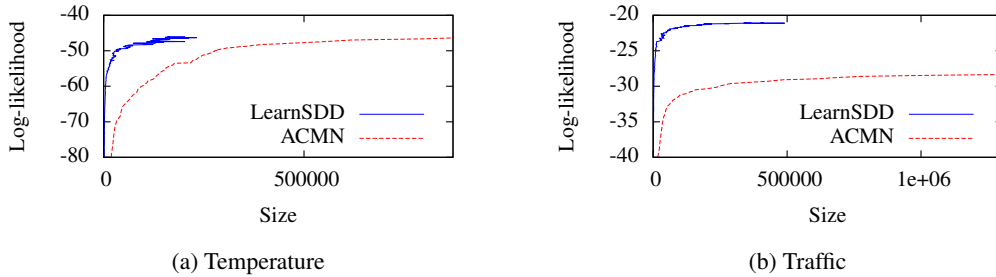

(a) Temperature            (b) Traffic

Figure 2: The size and log-likelihood of the models learned by LearnSDD and ACMN. Ideally, the model is small with high accuracy (upper left corner), which is best approached by the LearnSDD models.

334 out of 388 of the query settings whereas this only happens in 25 settings for LearnSDD. The SDD can answer all parity questions and positive word queries in less than three hundred milliseconds and the group comparison in less than three seconds. It can answer the voting query with up to 75% of the votes unknown in less than ten minutes. These results demonstrate LearnSDD's superior ability to answer complex queries compared to ACMN and allow us to positively answer **(Q2)**.

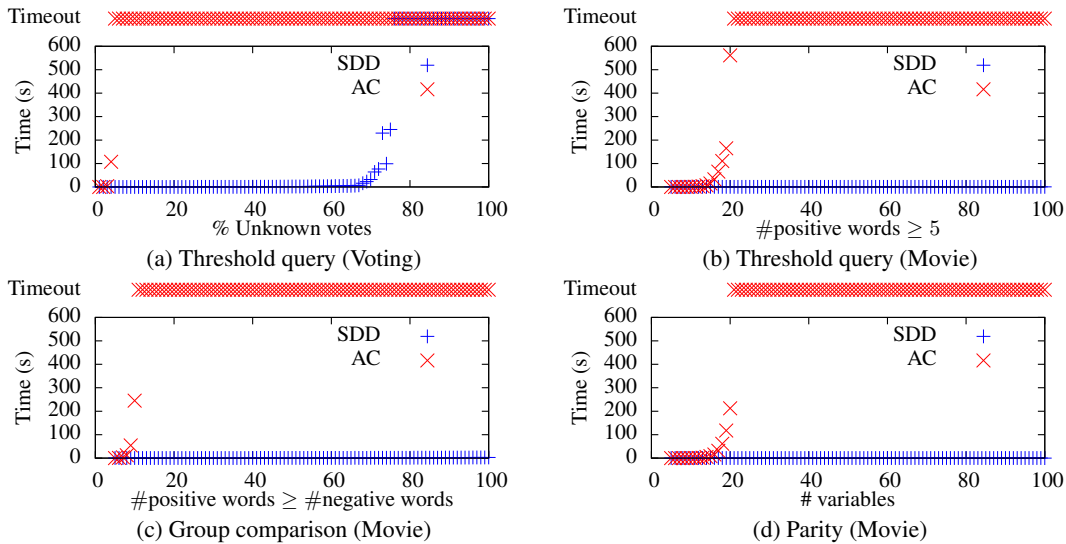

(a) Threshold query (Voting)            (b) Threshold query (Movie)

(c) Group comparison (Movie)            (d) Parity (Movie)

Figure 3: The time for SDDs vs. ACs to answer complex queries, varying the number of query variables. SDDs need less time in all settings, answering nearly all queries. ACs timeout in more than 85% of the cases.

## 6   Conclusions

This paper highlighted the fact that tractable learning approaches learn models for only a restricted classes of queries, primarily focusing on the efficient computation of conditional probabilities. We focused on enabling efficient inference for complex queries. To achieve this, we proposed using SDDs as the target representation for tractable learning. We provided an algorithm for simultaneously learning a Markov network and its SDD representation. We proved that SDDs support polytime inference for complex symmetric probability queries. Empirically, SDDs enable significantly faster inference times than ACs for multiple complex queries. Probabilistic SDDs are a closely related representation: they also support complex queries (in structured probability spaces) [26, 27], but they lack general-purpose structure learning algorithms (a subject of future work).

### Acknowledgments

We thank Songbai Yan for prior collaborations on related projects. JB is supported by IWT (SB/141744). JD is partially supported by the Research Fund KU Leuven (OT/11/051, C22/15/015), EU FP7 Marie Curie CIG (#294068), IWT (SBO-HYMOP) and FWO-Vlaanderen (G.0356.12). AC and AD are partially supported by NSF (#IIS-1514253) and ONR (#N00014-12-1-0423).

## Footnotes

[1] The literature typically shows hardness for polytrees. Results carry over because these have compact ACs.

[2]https://dtai.cs.kuleuven.be/software/learnsdd

[3]`http://reasoning.cs.ucla.edu/sdd/`

[4]`http://www.cs.cornell.edu/people/pabo/movie-review-data/`

[5]`http://tartarus.org/martin/PorterStemmer/` and `http://scikit-learn.org/`

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
