[Supplementary Material]

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

[6]ACE is available at `http://reasoning.cs.ucla.edu/ace/`, which uses C2D to compile CNFs to d-DNNFs, which is also available at `http://reasoning.cs.ucla.edu/c2d/`.

[7]The SDD package is available at `http://reasoning.cs.ucla.edu/sdd/`.

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

# A   Sentential Decision Diagrams

The Sentential Decision Diagram (SDD) is a newly introduced target representation for propositional knowledge bases [23, 28–31]. It is a strict subset of deterministic, decomposable negation normal form (d-DNNF), which has been used as a target representation for probabilistic graphical models, such as Markov networks and Bayesian networks [8, 6]. The ACE system is a state-of-the-art system for probabilistic inference, which is based on encoding a network as a propositional knowledge base (in CNF), which in turn is compiled into d-DNNF.[6] SDDs and d-DNNFs impose two properties on propositional knowledge bases, decomposability and determinism, that enable the tractability of probabilistic (and logical) inference queries [32]. For example, decomposability asserts that the branches of a conjunction have sets of variables that are pairwise disjoint; this enables (pure) MAP queries in Markov networks (and satisfiability queries in propositional knowledge bases).

Figure 4a depicts an SDD: paired-boxes $\boxed{p \mid s}$ are called *elements* and represent conjunctions ($p \wedge s$), where $p$ is called a *prime* and $s$ is called a *sub*. Circles are called *decision nodes* and represent disjunctions of the corresponding elements. SDDs satisfy stronger properties than d-DNNFs, allowing one, for example, to conjoin or disjoin two SDDs in polytime. In contrast, this is not possible in general with d-DNNFs (corresponding to ACs) [32], and other tractable representations for probabilistic graphical models. As we describe later, the ability to conjoin and disjoin SDDs efficiently is critical for the incremental learning of Markov networks.

An SDD is constructed for a given *vtree,* which is a full binary tree whose leaves are in one-to-one correspondence with the given variables; see Figure 4b. The SDD is canonical for a given vtree (under some conditions) and its size depends critically on the vtree used. Ordered Binary Decision Diagrams (OBDDs) [33] are a strict subset of SDDs: OBDDs correspond precisely to SDDs that are constructed using a special type of vtree, called a right-linear vtree [23]. Theoretically, SDDs come with size upper bounds (based on treewidth) [23] that are tighter than the size upper bounds that OBDDs come with (based on pathwidth) [34–36]. In practice, dynamic compilation algorithms can find SDDs that are orders-of-magnitude more succinct than those found using OBDDs. Such algorithms are enabled by the canonicity of SDDs, which allows one to search the space of vtrees

Figure 5: Vtree for Figure 1

to find succinct SDDs. As we shall describe later, the ability to dynamically minimize an SDD is critical for learning tractable representations of Markov networks.

Every decision node in an SDD is *normalized* for some vtree node. In Figure 4a, each decision node is labeled with the vtree node it is normalized for. Consider a decision node with elements $\boxed{p_1 \mid s_1}, \ldots, \boxed{p_n \mid s_n}$, and suppose that it is normalized for a vtree node $v$ which has variables $\mathbf{X}$ in its left subtree and variables $\mathbf{Y}$ in its right subtree. We are then guaranteed that each prime $p_i$ will only mention variables in $\mathbf{X}$ and that each sub $s_i$ will only mention variables in $\mathbf{Y}$ (this ensures decomposability). Moreover, the primes are guaranteed to represent propositional sentences that are consistent, mutually exclusive, and exhaustive (this ensures determinism). For example, the top decision node in Figure 4a has elements that represent the following sentences:

$$\{(\underbrace{A \wedge B}_{\text{prime}}, \underbrace{\text{true}}_{\text{sub}}), (\underbrace{\neg A \wedge B}_{\text{prime}}, \underbrace{C}_{\text{sub}}), (\underbrace{\neg B}_{\text{prime}}, \underbrace{D \wedge C}_{\text{sub}})\}$$

One can verify that these primes and subs satisfy the properties above.

In our experiments, we use the SDD package provided by the Automated Reasoning Group at UCLA.[7] This package allows one to efficiently conjoin, disjoin and negate SDDs, in addition to computing weighted model counts (i.e., the partition function of a Markov network and its derivatives) in linear time [8, 37]. Moreover, this package supports the dynamic minimization of SDDs, by searching the space of vtrees [29], which is critical for controlling the complexity of inference in an SDD representation of a Markov network.

## B    Weighted Model Counting

We review here how to reduce inference in Markov networks to weighted model counting [6]. Figure 1a in the main text highlights a simple Markov network over two binary variables $A$ and $B$; we note that the reduction to weighted model counting also generalizes to multi-valued variables in a straightforward way.

We can encode a Markov network as a propositional knowledge base $\Delta$ whose weighted model count will correspond to the partition function of the Markov network: $Z = \sum_x \exp\left(\sum_j w_j f_j(x)\right)$. We first define the propositional variables of the knowledge base $\Delta$. First, for each Markov network variable $X_i$ we define *indicator variables* $I_{X_i}$ of $\Delta$. Second, for each feature $f_j$ with weight $w_j$, we define a *parameter variables* $P_j$. In Figure 1a, we have two binary features, with their corresponding parameter variables in the knowledge base $\Delta$.

Our knowledge base $\Delta$ is composed of certain constraints, one for each feature $f_j$ in the Markov network. Assuming each feature is a term (a conjunction of literals), then we denote $\alpha_j$ as the corresponding term over indicator variables $I_{X_i}$. Our knowledge base $\Delta$ is then composed of the constraints $\alpha_j \Leftrightarrow P_j$ for each feature $f_j$. In our example, we include four constraints in the knowledge base $\Delta$, one for each feature. For the two binary features of Figure 1a, we introduce two constraints: for the term $A \wedge B$ we introduce the constraint $P_1 \Leftrightarrow (I_A \wedge I_B)$ and for the term $\neg A \wedge \neg B$ we introduce the constraint $P_2 \Leftrightarrow (\neg I_A \wedge \neg I_B)$. Figure 1b highlights an SDD representation of this example, and the corresponding vtree in Figure 5.

To perform weighted model counting, we need to specify weights on each literal of the knowledge base $\Delta$. For each indicator variable $X_i$, we set both literal weights $W(I_{X_i})$ and $W(\neg I_{X_i})$ to one. For each parameter variable, we set the positive literal weight $W(P_j)$ to the weight of the corresponding feature, i.e., $W(P_j) = \exp(w_j)$. We further set the negative literal weight $W(\neg P_j)$ to one. The models $w$ of the resulting knowledge base $\Delta$ are now in one-to-one correspondence with rows of the joint distribution table induced by our Markov network. In particular, consider the weight $W(w)$ of a model $w$, and the weighted model count $W(\Delta)$ of our knowledge base $\Delta$:

$$W(w) = \prod_{w \models \ell} W(\ell) \qquad W(\Delta) = \sum_{w \models \Delta} W(w)$$

Note that if model $w$ is consistent with term $\alpha_j$ of feature $f_j$, then the corresponding parameter variable $P_j$ is set to true in model $w$, and the model weight $W(w)$ includes the feature weight $w_j$. For example, we have the following model $w$ and model weight $W(w)$:

$$w = (I_A, I_B, P_1, \neg P_2)$$
$$W(w) = W(P_1)$$
$$= \exp(w_1) = \Pr(A, B) \cdot Z.$$

Further, the weighted model count yields the partition function $Z$. We incorporate evidence by setting to zero the weights of any indicator variable $I_{X_i}$ that is not compatible with the evidence. The weighted model count then corresponds to the probability of evidence (after normalization by the partition function).

# C    Protocols for SDD Manipulation

This section provides insight in how to manipulate SDDs for learning and answering queries. First we show how to add a feature to an existing model and then we show how to use an SDD to answer complex probability queries with evidence.

## C.1    Adding Features

**Given:**    Markov network $M$ with a corresponding SDD $S_M$ and a feature $f_i$ with corresponding weight $w_i$. Feature $f_i$ is a logical formula over variables or previously added features, e.g. $a \wedge f_j$, where $a$ is a variable and $f_i$ a previously added feature.

**Construct:**    A new SDD $S_{M'}$ that corresponds to the model $M'$ which is the combination of the original model $M$ and the new feature $f_i$

**Protocol:**

1. Introduce a new parameter variable $P_{f_i}$.
2. Compile the formula $P_{f_i} \Leftrightarrow f_i$ (e.g. $P_{f_i} \Leftrightarrow a \wedge f_j$) to SDD $S_{f_i}$.
3. Add the feature to the model by conjoining the two SDDs: $S_{M'} = \mathrm{conjoin}(S_M, S_{f_i})$
4. During weighted model counting, the new variable $P_{f_i}$ will have $w_i$ as a positive weight and 1 as a negative weight.

## C.2    Answering Queries

**Given:**    Markov network $M$ with a corresponding SDD $S_M$ and evidence $e$ where $e$ is a subset of the variables with their given values, e.g. $e = \{E = 1, F = 0\}$.

**Answer:**    A query $\Pr(q|e, M)$, e.g. threshold query $\Pr(\#(A, B, C, D) > 1|e, M)$.

**Protocol:**

1. Compile $q$ to SDD $S_q$.

   As an example, consider the following formula for the query $\#(A, B, C, D) > 1$:

   $$\alpha_{\#(A,B,C,D)>1} = (A \wedge \alpha_{\#(B,C,D)>0}) \vee (\neg A \wedge \alpha_{\#(B,C,D)>1})$$

   which has the following sub-formula

   $$\alpha_{\#(B,C,D)>1} = (B \wedge \alpha_{\#(C,D)>0}) \vee (\neg B \wedge \alpha_{\#(C,D)>1})$$
   $$\alpha_{\#(C,D)>1} = (C \wedge \alpha_{\#(D)>0}) \vee (\neg C \wedge \alpha_{\#(D)>1})$$
   $$\alpha_{\#(D)>1} = \bot$$

   $$\alpha_{\#(B,C,D)>0} = (B \wedge \top) \vee (\neg B \wedge \alpha_{\#(C,D)>0})$$
   $$\alpha_{\#(C,D)>0} = (C \wedge \top) \vee (\neg C \wedge \alpha_{\#(D)>0})$$
   $$\alpha_{\#(D)>0} = D$$

   The above formulas correspond to sub-SDDs (each corresponds to a decision node, except for $\alpha_{\#(D)>1}$ and $\alpha_{\#(D)>0}$ which correspond to terminal nodes). These formulas can be constructed recursively using the apply operation for SDDs, to conjoin SDD literals and SDD sub-formula. We also observe that the sub-formulas $\alpha_{\#(C,D)>0}$ and $\alpha_{\#(D)>0}$ appear multiple times, and can be re-used via caching.

2. Set the evidence $e$ in SDD $S_M$ by conditioning on the values, this results in a new SDD $S_{M,e}$. As a bonus, this step will reduce the size of the SDD and therefore speed up the next steps.

3. If the query also contains evidence variables, condition the $S_q$ on the evidence values as well. This results in a new SDD $S_{q,e}$

4. Use SDD $S_{M,e}$ to calculate $WMC_{M,e}$

5. Conjoin the model and the query $S_{M \wedge q,e} = \text{conjoin}(S_{M,e}, S_{q,e})$

6. Use SDD $S_{M \wedge q,e}$ to calculate $WMC_{M \wedge q,e}$

7. The answer to the query is

$$\Pr(q|e, M) = \frac{WMC_{M \wedge q,e}}{WMC_{M,e}}$$

## D  Proof of Theorem 1

*Proof.* First, we have $\Pr_M(q) = \frac{WMC(q \wedge M)}{WMC(M)}$. We can compute the weighted model count of network $M$, $WMC(M)$, in time $O(m)$ given an SDD for network $M$ of size $m$. Thus, it suffices to show that we can compute $WMC(q \wedge M)$ in time $O(mn^3)$. First, any symmetric function $q$ has a structured d-DNNF of size $O(n^3)$, for any vtree [38]. Second, conjoining two structured d-DNNFs, with sizes $m_1$ and $m_2$, takes time $O(m_1 m_2)$, when they share the same vtree [38]. Since any SDD is a structured d-DNNF, conjoining an SDD of size $m$ with any symmetric function takes time $O(mn^3)$, and results in a structured d-DNNF of size $O(mn^3)$. Since we can also perform weighted model-counting on a structured d-DNNF in time linear in its size, we can compute $WMC(q \wedge M)$ in time $O(mn^3)$. Similarly, for the special cases of parity, $k$-threshold, and exactly-$k$ functions, where the number of structured d-DNNF nodes respecting each vtree node can be bounded by a function of $k$ [38]. $\square$

## E  Complex Query Experiments

### E.1  Time Needed by Simpler and Harder Queries

To contrast the symmetric queries with simpler and harder cases, we also evaluated simple conjunctive queries and weighted $k$-threshold queries using the movie data set. We vary the number

of variables involved in the query from 5 to 100 variables. Each literal in the weighted $k$-threshold query has a corresponding weight an the total weight of true literals must be bigger than some threshold. For the experiment, all the variables were assigned random weights and the threshold was set to half of the maximum total weight.

Figure 6 shows the time needed to answer the conjunctive and weighted $k$-threshold query. As expected, both the AC and SDD can answer the conjunctive queries efficiently: they always need under 0.1 seconds. They both have difficulties with the weighted $k$-threshold query, but the SDD does outperform the AC. SDDs are not guaranteed to efficiently answer weighted $k$-threshold queries, but in practice they often do. Even when Figure 6b shows a timeout, the SDD was able to answer the query efficiently for most test examples, but at least one example exceeded the time limit. Thus even though we cannot guarantee efficient inference in all cases, it will still often be beneficial to use SDDs as the underlying tractable representation.

(a) Conjunctive queries (Movie)      (b) Weighted threshold (Movie)

Figure 6: The time needed by an SDD vs. an AC to answer a conjunctive and a weighted $k$-threshold query, for various number of variables in the query. The SDD and AC perform similarly for the conjunctive query and the SDD outperform the AC for the weighted $k$-threshold query.

## E.2 Conditional Log-Likelihoods

Using the training set, we learned models with both LearnSDD and ACMN. For LearnSDD, we tried setting $\alpha$ to 1.0, 0.1, 0.01 and 0.001. For ACMN, we tried setting the per-split penalty $ps$ to 2, 5 and 10. Because the primary goal of experiment was to evaluate the inference time, the learned AC and SDD for each data set needed to be of similar size. To ensure this, we first picked the SDD with the best log-likelihood on the validation data. Subsequently, we selected the AC with the best validation set log-likelihood that was smaller than the chosen SDD. We picked an SDD first because the largest SDD for each setting was smaller than the largest AC for each setting.

Figure 7 shows the per example average conditional log-likelihood for the complex queries. The conditional log-likelihood expresses how close a query's predicted answer is to its actual answer according to the data. The higher the log-likelihood of the model, the better it is at answering the query.

The SDD gives better results for the group query in Figure 7c, in all the other settings, the results are comparable.

Figure 7: Conditional log-likelihood of the answers to the complex queries using an SDD vs. an AC, for various number of variables in the query. The SDD and AC result in most cases in similar conditional log-likelihoods.