[Reviews · NeurIPS 2015]

Submitted by Assigned_Reviewer_1

"Tractable Learning for Complex Probability Queries" recommends the use of Sentential Decision Diagrams, instead of arithmetic circuits, for learning and querying Markov networks.

For learning they propose a greedy algorithm that adds edges when a score which is a penalized log-likelihood is increased.

In the main theorem, they demonstrate that when the resulting learned SDDs are small, it is possible to tractably query symmetric (permutation invariant) functions.

The experimental results show that SDDs are more accurate than ACs of the same size and that the queries are significantly faster.

This paper is well written, and the point of the paper (which is the recommendation of using SDDs over ACs) is convincingly argued.

Because I am not an expert in this field, I cannot speak to the novelty or importance of this result.

But the main result seems correct and the experiments are very convincing.

To me it seems like an important result that is novel.

The only typo that I found was line 256 "with with".
Summary: I am not an expert in this field so I cannot speak to the novelty or importance of this work, but the paper is very well written, the main result is correct, and the experiments are convincing.

Submitted by Assigned_Reviewer_2

This paper presents a tractable learning method, called LearnSDD, for dealing with complex probability queries. Built on Sentential Decision Diagrams (SDDs), this method learns a Markov network and its circuit representation. Also, the authors proved that, the proposed method takes polynomial time for some class of complex probability queries.

This paper may present some interesting ideas, but it is really hard to follow, maybe due to organization issues and lacks of some details. For example, some backgrounds, such as "what are complex queries", connections of graphical models to "queries" and related studies, are not given. In Section 2.1, the authors presented Markov networks, but what is the connection to the presented method?

Moreover, the motivations of the main algorithm is not clear. The proposed LearnSDD method, namely Algorithm 1, is built upon SDDs. But from current paper, I am not clear about the motivations. The validity of this algorithm is not clear.

Moreover, the score function in (1) is also confusing. Without necessary details, I do not understand why it is in current form. Also, how to compute Pr(T|M)?

For the experiments, I think the comparison so far is not sufficient. For example, there is only one comparison method, which is not enough.

The first argument in the paper, i.e. "Tractable learning is a new machine learning paradigm", is not correct. We may say "Tractable structural learning". In fact, "Tractable" just implies that certain operations are efficient, but it is not a machine learning paradigm.

Summary: This paper may present some interesting ideas, but it is hard to follow. Moreover, motivations of the main algorithm is not clear, and some essential details are not given.

Submitted by Assigned_Reviewer_3

Abstract: the paper introduces LearnSDD an algorithm that learns log-linear models for discrete random variables but adds a penalty term for models that are expensive at query time. Compared to earlier work in this direction the paper studies a new way of describing models (SDDs instead of ACs) and is interested in "complex queries", e.g. counts of groups of binary instead of MAP, or marginals. The computational complexity of complex queries are not directly addressed in the algorithm, but as it turns out the choice of SDD as model space also has good run-time performance for certain complex queries (Theorem 1).

Quality: there are no obvious errors, but some definitions in the proof are missing. Some key elements in the algorithm are not motivated/discussed (see comments below)

Clarity: The presentation is good enough, but can be improved. Computational complexity is key in the paper but discussed late and only mentioned not derived.

Originality: the paper provides an addition following papers in a workshop at icml 2014.

Significance: the argument and examples that "complex queries" are important are mildly convincing. Not many people will be impacted by this. Also the proposed algorithm has this only as a by-product.

Comments:

- Other than the empirical results, is there a motivation for the SDD being a good/suitable model class? Together with the order of features and score (i.e. the implied prior over models)?

- The order in which features are proposed is crucial in the greedy algorithm. Why is this order suitable?

- In the discussion of the score (Sec 3.1) it is nice to discuss the computational complexity of the queries as a function of |SDD|.

- The score currently is a weighted combination of the relative increase in computational cost but absolute cost in log likelihood. Is this deliberate?

- Given that Theorem 1 is one of the main claimed contributions of the paper, and the proof is only a few lines, it would be better to add it to the paper itself (you even have space for it). What is WMC in the proof?

Summary: The paper adds to a recent series of papers on learning models while penalizing query time computational complexity. Not all key decisions in the algorithm are perfectly explained and the argument for looking at "complex queries" (e.g. counts of binary variables instead of MAP, marginals) is somewhat contrived.

Submitted by Assigned_Reviewer_4

The paper presents an algorithm for learning a Markov network together with its underlying representation of Sentential Decision Diagrams. By using SDD, it is argued in the paper that more complex queries than usual MPE, such as counting, threshold, comparison, etc become tractable. The algorithm is built from a previous learning algorithm, namely the ACMN learner, which uses arithmetic circuits instead of SDDs to allow tractable queries in Markov networks. Experiments are conducted in four datasets, where the new LearnSDD is compared to ACMN.

The paper address an important research topic, it is generally clear and well-written, and besides empirical results it also presents theoretical results concerning the complexity of performing queries in Markov networks represented as SDDs. The questions that arose about the paper are the following.

- How the representation and answering mechanism addressed in the paper are compared to probabilistic SDD, by D. Kisa et al., 2014? Also, there are a number of previous representations that also deals with complex queries, such as the ones that are able to deal with aggregation functions in SRL. Would it be possible to compare (theoretically and/or empirically) the answering machinery of MNs as SDDs with those type of representations? I believe the paper lacks

a related work section, where those comparisons could be discussed.

- Is it possible to learn the weights efficiently even in the presence of missing data?

- The compilation of the SDD for a query together with the underlying SDD for the MN it is a key step for answering complex queries without having to create multiple conjunctive queries (as it is done in ACs). Therefore, a thorough explanation should also be given about this step in the paper, maybe with an example. Also, the relationship of this compilation step with mutex features should be discussed in the paper.

Minors: -remove the second "it" in "after it was learned it" -remove second "with" in 4.1, second paragraph -the alpha parameter should be given as input to the score function in the algorithm -as the algorithm 1 also terminates when it runs out of time (according to the explanation in the text), this stopping requirement should also appear in the algorithm. -The highest scoring feature is only added to the current model if the score

of the model plus the new feature is higher than before (according to the algorithm). This should be also said in the text (and not only saying that the feature with the highest score is permanently added to the model).
Summary: Although this is a good paper, addressing the important topic of tractability of complex and

(because of their complexity) not usual queries, it could have presented and compared the devised approach with other related work, instead of only ACMN, such as probabilistic SDD. Also,

the representation of complex queries within the learned model could be clearer in the paper, as this is the main goal of the approach.

Submitted by Assigned_Reviewer_5

Summary of paper: The paper deals with learning SDDs (from previous work) that can be used to infer complex queries. Inference for these type of queries could not be done efficiently in previous models like the ACMNs. The paper provides details about the learning algorithm (which is similar to the ACMN learning algorithm based on greedy selection), different types of complex queries (symmetric and asymmetric that can be converted to symmetric), and experimental evaluation on real data to answer how well SDDs do and how fast they are. Complexity bounds on the inference complexity are also provided.

Quality: The paper is of high quality and is well written. Did not find any errors in text or any equations.

Clarity: Along with the supplemental text (description of SDDs and WMC), the paper is quite clear to follow. Suggestion but not required would be to provide similar summary text in the supplemental material for ACMN especially since you have the figure in 1(c).

Originality: To the best of my knowledge, this is a novel method to learn SDDs from queries and trying to infer more difficult queries than was previously done using tractable models.

Significance: Given that the paper provides another step in the direction of use of tractable models to solve even more difficult queries, there could be other ideas that could build on what is provided here.

Questions, clarifications and suggestions to improve paper:

1) Are sum-product networks any better than ACMNs to solve these complex queries? If so, how would they compare against use of SDDs ?

2) You use the work conjoin at multiple places, it might be help define the operator.

3) In the supplementary text you mention additional unary parameters w_A and w_B ? Is there any reason they are not included in the tree 1b and 1c ? Are they not needed to be included in the SDD ?

4) In Algorithm 1, you pass \alpha as a parameter. You should pass \alpha as a parameter to score(M, T) since score depends on it.

5) What step of the algorithm are you doing dynamic SDD minimization ? Is it done in M.add(f) step ?

6) Is it possible that the minimization or compressing can reduce the size of the SDD more in step t+2 by selecting features f1 and f2 combined compared to when you would have selected f3 and f4 sequentially in a greedy way ? i.e. does selecting features in a greedy fashion cause generating a SDD that might not be optimally minimized ?

7) It would be good to have example queries where ACMN does similar to SDDs as well as asymmetric queries that are not tractable by SDDs for completeness.

8) It might be better to use func() instead of # in Table 1.

9) Fix opening double quotes on page 7. ... vote "yes" on a bill ...

10) In the positive, negative movie review question, is there any reason you don't use all characters instead of restricting to first 750 characters ? does it affect performance and how ?

11) While answering the second question posed and using complex queries on the voting and movie dataset, is there any reason you do not look at the accuracy as well as the time especially for queries that do not time out. It seemed like the Mutex queries and Complex queries you had are quite different and you didn't test for the accuracy in terms of log-likelihood on the complex queries.

12) Complete the references with details (i.e. for example put in page numbers for references 3, 6, 8, 12 etc)
Summary: The paper provides an interesting new direction to learn a MN structure (SDD) and infer complex probability queries which could not be inferred efficiently using previous MN models.

Submitted by Assigned_Reviewer_6

The paper proposes an new algorithm for structure learning of binary MRF networks; the core idea is using Sequential Decision Diagrams (SDD) as compact representation of MRF; it allows more flexibility to control structure complexity and more efficient computation for hard probability queries, than traditional arithmetic circuits (AC) representations; the work is interesting.

It is slight disappointing that the idea is still embedded in heuristic search procedure; an improvement can be to encode the SDD creation or operations in a stochastic search procedure, namely a MCMC framework, to obtain better learning results.

Summary: The paper proposes an new algorithm for MRF structure learning by introducing a compact representation from propositional logic, i.e., Sequential Decision Diagrams (SDD); it is more flexible and efficient than arithmetic circuits representation; it is an interesting work.

Author Feedback
Author rebuttal: We thank the reviewers for their valuable feedback. We will address the suggestions for a final version. Now we respond to each reviewer.

REVIEWER 2

We will add a discussion on PSDDs and SRL:

There does not currently exist a general structure learning algorithm for PSDDs. PSDD learning (Choi IJCAI15) requires a structured probability space (logical constraints) that is compiled into SDD and parametrized during learning. Such logical constraints are absent from the general learning setting in the present work. PSDDs do inherit from SDDs the ability to efficiently do complex queries.

Many SRL approaches learn aggregates or combining functions (eg Getoor IJCAI01, Vens ILP04, Natajaran AMAI08). Our work differs in that we
1) learn propositionally where aggregates are over a single row in the data (SRL aggregates are over multiple rows), and
2) guarantee tractable inference (SRL representations usually do not).

We will add an example of SDD compilation to the supplement.

REVIEWER 3

1. Since SPNs and ACs are interchangeable (Rooshenas and Lowd 2014, Propositions 1&2), SPNs are also not efficient for complex queries.

3. This is a typo. We refer to an old version of Fig 1.

5. Minimization is done after M' <- M.add(f) and after M <- M_best.

6. Yes. Both structure learning and SDD minimization are NP-hard, so both are done with a best-effort greedy-search approach.

7. SDDs and ACs perform similarly on conjunctive queries. Our models are intractable for sentences without a compact SDD, such has hard satisfiability problems. We will add a weighted k-threshold example.

10. We consider an active sensing setting to predict features of the review without reading it entirely. If the entire review is observed, there is no need to predict these features as they can be computed directly.

11. We will add conditional log-likelihoods (CLL) of complex queries. A preliminary test for voting gives results in favor of SDDs: for 1-4% unknown votes (where AC are tractable), the AC CLLs are -0.47,-0.71,-0.83, -0.73, while SDD gets -0.47,-0.58,-0.79, -0.59

REVIEWER 4

1. What are complex queries

This is explained in Section 4, lines 226-238.

2. Connection of graphical models to queries

Answering queries (inference) is the main tasks in graphical models (line 94)

3. Connection to Markov networks

The paper is about learning MNs that allow efficient inference. SDDs and ACs are tractable representations of MNs (line 110).

4. Motivations of the main algorithm are unclear

Using SDDs is motivated in lines 64-67, 158-161 and 191-193.

5. "Tractable learning is a new ... paradigm": not correct

We use tractable learning to refer to the recent trend of learning general-purpose tractable probabilistic models with complex dependency structures. While there has been work in this direction since the 60s, the paradigm has resurged in recent years, with lots of activity and the first ICML workshop on the topic in 2014.

REVIEWER 6

1. MCMC vs heuristic search

This is an interesting future direction. Your suggestion highlights a benefit of SDDs: they allow a MCMC-style search because it is possible to efficiently add and remove features. In contrast, removing a feature from ACs would require recompiling the AC from scratch, which would be very computationally expensive.

We used a greedy top-down search to make the search as close as possible to the ACMN baseline. Thus we can see if the benefit comes from using SDDs and not from a different search algorithm.

REVIEWER 7

1. Why complex queries

Complex properties such as counts and threshold appear naturally in related domains such as neural networks and relational learning (cf. Reviewer 2). We believe they have not been used to their full potential in the graphical model world because of their intractability, which we aim to resolve.

2. Motivate SDD as a good model class

SDDs are useful for tractable learning because 1) they give a direct measure for the complexity of inference, 2) they are more flexible than other representations (ACs and SPNs) because we can add any kind of feature and perform more queries efficiently and 3) with minimization one can reduce the size of an SDD during learning.

3. Importance of feature order

The order is not crucial because in each iteration, the algorithm considers all the features and then only adds the best. If no minimization is performed along the way, the algorithm is even oblivious to the order in which the features are presented.

4. Structure of score function

We take the relative increase in size because the worst case size increase is linear in the current model size. The size increase will thus always get worse when the model grows bigger. When using absolute size increase, from a certain point it only adds features based on size increase but they no longer contribute to likelihood.

5. Theorem 1 proof

We will add a proof outline to the paper.